# The Ser290Asn and Thr715Pro Polymorphisms of the *SELP* Gene Are Associated with A Lower Risk of Developing Acute Coronary Syndrome and Low Soluble P-Selectin Levels in A Mexican Population [note 1]

**DOI:** 10.3390/biom10020270

**Published:** 2020-02-11

**Authors:** Gabriel Herrera-Maya, Gilberto Vargas-Alarcón, Oscar Pérez-Méndez, Rosalinda Posadas-Sánchez, Felipe Masso, Teresa Juárez-Cedillo, Galileo Escobedo, Andros Vázquez-Montero, José Manuel Fragoso

**Affiliations:** 1Department of Molecular Biology, Instituto Nacional de Cardiología Ignacio Chávez, Mexico City 14080, Mexico; mayadermata@ciencias.unam.mx (G.H.-M.); gvargas63@yahoo.com (G.V.-A.); koapa_93and@hotmail.com (A.V.-M.); 2Department of Endocrinology, Instituto Nacional de Cardiología Ignacio Chávez, Mexico City 14080, Mexico; rossy_posadas_s@yahoo.it; 3Laboratory of Translational Medicine, UNAM-INC Research Unit, Instituto Nacional de Cardiología, Ignacio Chávez, Mexico City 14080, Mexico; f_masso@yahoo.com; 4Commissioned of the Research Unit in Clinical Epidemiology, Hospital Regional No. 1, Dr. Carlos McGregor Sánchez Navarro, Instituto Mexicano del Seguro Social, Mexico City 14080, Mexico; terezillo@exalumno.unam.mx; 5Unit of the Experimental Medicine, Hospital General de Mexico, Dr. Eduardo Liceaga, Mexico City 14080, Mexico; gescobedog@msn.com

**Keywords:** acute coronary syndrome, P-selectin, genetics, polymorphisms, susceptibility

## Abstract

Recent studies have shown that P-selectin promotes the early formation of atherosclerotic plaque. The aim of the present study was to evaluate whether the *SELP* gene single nucleotide polymorphisms (SNPs) are associated with presence of acute coronary syndrome (ACS) and with plasma P-selectin levels in a case-control association study. The sample size was estimated for a statistical power of 80%. We genotyped three *SELP* (*SELP* Ser290Asn, *SELP* Leu599Val, and *SELP* Thr715Pro) SNPs using 5’ exonuclease TaqMan assays in 625 patients with ACS and 700 healthy controls. The associations were evaluated with logistic regressions under the co-dominant, dominant, recessive, over-dominant and additive inheritance models. The genotype contribution to the plasma P-selectin levels was evaluated by a Student’s t-test. Under different models, the *SELP* Ser290Asn (OR = 0.59, *pC_Co-_*_Dominant_ = 0.047; OR = 0.59, *pC*_Dominant_ = 0.014; OR = 0.58, *pC_Over-_*_Dominant_ = 0.061, and OR = 0.62, *pC*_Additive_ = 0.015) and *SELP* Thr715Pro (OR = 0.61, *pC*_Dominant_ = 0.028; OR = 0.63, *pC_Over-_*_Dominant_ = 0.044, and OR = 0.62, *pC*_Additive_ = 0.023) SNPs were associated with a lower risk of ACS. In addition, these SNPs were associated with low plasma P-selectin levels. In summary, this study established that the *SELP* Ser290Asn and *SELP* Thr715Pro SNPs are associated with a lower risk of developing ACS and with decreased P-selectin levels in plasma in a Mexican population.

## 1. Introduction

Acute coronary syndrome (ACS) comprises a spectrum of obstructive coronary artery diseases that most commonly arise from plaque rupture and/or erosion, leaving the vulnerable lipid-rich core exposed to the circulation. As a result, platelets and the coagulation cascade are activated, leading to acute thrombotic occlusion [1,2]. This syndrome is a consequence of atherosclerosis associated with a strong inflammatory component, which is immune mediated by chemokines. These molecules have an important role in the development of atherosclerotic plaque [3,4,5]. P-selectin is a chemokine, which mediates lymphocyte and monocyte recruitment, rolling, and diapedesis to the areas of inflammation [4,5,6]. Experimental studies have shown that higher expression of SELP increases adhesion, monocytes rolling to the vascular wall, accumulation of oxidized low-density lipoproteins, and the early formation of atherosclerotic plaque and other inflammatory diseases [4,5,6,7]. 

P-selectin contains 17 exons and is encoded by the *SELP* gene located on chromosome 1q21-q24 spanning <50 kb [8]. Recently, three single nucleotide polymorphism (SNPs) in the *SELP* gene in the exons 7, 12, and 13 [positions *G1057A* Ser290Asn (rs6131), *G1980T* Leu599Val (rs6133), and *A2331C* Thr715Pro (rs6136)] have been associated with myocardial infarction, hypertension, coronary heart disease, lupus erythematosus, type 2 diabetes mellitus (T2DM), and atherosclerosis [8,9,10,11,12,13]. Nonetheless, the association between these SNPs and other inflammatory diseases, such as diabetic retinopathy and multiple sclerosis is controversial, with negative results [14,15]. 

Considering the prominent role of P-selectin as a key in the chain of events leading to atherosclerotic plaque formation, the aim of this study was to investigate the association of three *SELP* SNPs (Ser290Asn, Leu599Val and Thr715Pro) with the risk of developing ACS. Furthermore, we evaluated whether these SNPs were associated with plasma P-selectin levels in a Mexican population sample.

## 2. Subjects and Methods

### 2.1. Study Population 

This case-control study was carried out at the Instituto Nacional de Cardiologia Ignacio Chavez. The sample size was calculated for unmatched cases and controls with OpenEpi software (http://www.openepi.com/SampleSize/SSCC.html) with a statistical power of 80% and an alpha error of 0.05. Using this criterion, we included 625 patients with ACS (82% men and 18% women with a mean age of 57.97 ± 10.5 years) who were diagnosed based on clinical characteristics, electrocardiographic changes and biochemical markers of cardiac necrosis, according to guidelines from the European Society of Cardiology (ESC) and American College of Cardiology (ACC) [16,17]. The exclusion criteria were (1) patients with clear inflammatory pathologies on admission, such as infection established by clinical, laboratory, or image investigations, and (2) patients with an autoimmune disease or cancer previously diagnosed or documented during their hospitalization. Moreover, we included 700 healthy controls (66% men and 34% women with a mean age of 54.37 ± 7.65 years) coming from the Genetics of Atherosclerosis Disease (GEA) Mexican study previously described by Rosalinda-Posadas et al [18]. All healthy controls were asymptomatic and apparently healthy individuals without a family history of CAD and with a negative calcium score, indicative of the absence of subclinical atherosclerosis [18]. The exclusion criteria included not only the use anti-dyslipidemic, anti-hypertensive, and anti-diabetic drugs at the time of the study, but also congestive heart failure, as well as liver, renal, thyroid or oncological disease. All GEA participants were unrelated and of self-reported Mexican ancestry (3 generations). A Mexican mestizo was defined as a person who (1) was born in Mexico and (2) is a descendant of the original autochthonous inhabitants and of individuals (Caucasian and/or African, mainly Spaniards) who migrated to America in or after the XVI century. This study was conducted according to the principles of the Declaration of Helsinki and was approved by the Ethics and Research committee of our institution (registration number: 17CI09012010). Written informed consent was obtained from all individuals enrolled in the study.

### 2.2. Laboratory Analyses

After a 12-h overnight fast, EDTA blood samples were drawn and centrifuged within 15 min after collection; the plasma was separated into aliquots and immediately analyzed or frozen at −80 °C until analysis. Cholesterol and triglyceride plasma concentrations were determined by enzymatic/colorimetric assays (Randox Laboratories, UK). The phosphotungstic acid-Mg^2+^ method was used to determine HDL-C concentrations. LDL-C was estimated in samples with a triglyceride level lower than 400 mg/dl, using the modified Friedewald formula [19]. Plasma lipid concentrations were determined within 24 h after blood sample collection. We followed the National Cholesterol Education Project (NCEP) Adult Treatment Panel (ATP III) guidelines and thus defined dyslipidemia with the following levels: cholesterol > 200 mg/dl, LDL-C > 130 mg/dl, HDL-C < 40 mg/dl, and triglyceride > 150 mg/dl (http://www.nhlbi.nih.gov/guidelines/cholesterol/atp3_rpt.htm). Type 2 diabetes mellitus (T2DM) was defined with a fasting glucose ≥ 126 mg/dL and was also considered when participants reported glucose-lowering treatment or a physician diagnosis of T2DM. Hypertension was defined by a systolic blood pressure ≥ 140 mmHg and/or diastolic blood pressure ≥ 90 mmHg, or the use of oral antihypertensive therapy [18]. 

### 2.3. Genetic Analysis

DNA extraction was performed from peripheral blood in agreement with the method of Lahiri and Nurnberger [20]. The *SELP G1057A* Ser290Asn, *SELP G1980T* Leu599Val, and *SELP A2331C* Thr715Pro SNPs were genotyped using 5’ exonuclease TaqMan assays on a 7900HT Fast Real-Time PCR system according to manufacturer’s instructions (Applied Biosystems, foster City, CA, USA). In order to avoid genotyping errors, ten percent of the samples were determined twice; the results were concordant for all cases.

### 2.4. Determination of P-Selectin Levels 

Samples were aliquoted and stored at −70 °C for further use. Plasma P-selectin levels were measured using a quantitative sandwich enzyme immunoassay technique (ELISA) kit in accordance with the manufacturer’s instructions (Human P-Selectin/CD62P Quantikine ELISA Kit, R&D systems). The detection range was 0.8–50.00 ng/mL and the sensitivity was equal to the minimal detectable dose of this kit (≥ 0.121 ng/mL). 

### 2.5. Functional Prediction Analysis

Two in silico programs, the ESEfinder (http://rulai.cshl.edu/cgi-bin/tools/ESE3/esefinder.cgi?process=home) and SNP Function Prediction (http://snpinfo.niehs.nih.gov/cgi-bin/snpinfo/snpfunc.cgi) were used to predict the possible functional effect of the *SELP* SNPs. Both programs (ESEfinder2.0 and SNPinfo) analyzed the localization of the SNPs (e.g., 5’-upstream, 3’-untranslated regions, intronic) and their possible functional effects, such as amino acid changes in protein structure, transcription factor binding sites in promoter or intronic enhancer regions, and alternative splicing regulation by disrupting exonic splicing enhancers (*ESE*) or silencers [21,22].

### 2.6. Statistical Analysis

All statistical analysis in this study was performed using SPSS version 18.0 (SPSS, Chicago, Il). Data of continuous variables were expressed as median and percentiles (25th–75th), while data of discrete variables [e.g., frequency (n, %)] were analyzed using Chi-squared or Fisher’s exact tests. We used logistic regression tests to associate the SNPs with ACS under five inheritance models [16]. The correction of the p-values (pC) was performed with the Bonferroni test. Using the HAPLOVIEW version 4.1 software (Cambridge, MA, USA), we performed the haplotypes construction and linkage disequilibrium analysis (LD, D”). We tested whether our study population was in Hardy–Weinberg equilibrium (HWE) with a Chi-square test. Furthermore, we used the QUANTO software [http://biostats.usc.edu/software] to calculate the statistical power of our study and found it was 0.80. Using the Student’s t-test, we analyzed the contribution of the genotypes on the P-selectin plasma levels. The values were expressed as means ± SD. The level of significance was set at p < 0.05. 

## 3. Results

### 3.1. Characteristics of the Study Population

Clinical and biochemical characteristics of the ACS patients and healthy controls are shown in Table 1. There were significant differences between the ACS patients and healthy controls. Compared to healthy controls, the ACS patients had a higher frequency of T2DM, hypertension, dyslipidemia, and smoking habit. Conversely, the total cholesterol, triglycerides, and LDL-C levels in ACS patients were lower than those in the control group; this effect may be due to their treatment with statins. 

### 3.2. Allele and Genotype Frequencies

Genotype frequencies of the SNPs were in HWE. The frequencies of the *SELP Leu599Val* SNP was similar in ACS patients and healthy controls. Nonetheless, the SNPs [*SELP* Ser290Asn, and *SELP* Thr715Pro] were associated with a lower risk of ACS (Table 2). Under co-dominant, dominant, over-dominant, and additive models, the *A* (290Asn) allele of the *SELP* Ser290Asn SNP was associated with a lower risk of ACS (OR = 0.59, *pC_Co-_*_Dom_ = 0.047; OR = 0.59, *pC*_Dom_ = 0.014; OR = 0.58, *pC_Over-_*_Dom_ = 0.061, and OR = 0.62, *pC*_Add_ = 0.015, respectively). In the same way, under dominant, over-dominant, and additive models, the *C* (715Pro) allele of the *SELP* Thr715Pro SNP was associated with a lower risk of ACS (OR = 0.61, *pC*_Dom_ = 0.028; OR = 0.63, *pC_Over-_*_Dom_ = 0.044, and OR = 0.62, *pC*_Add_ = 0.023, respectively). All models were adjusted for gender, age, blood pressure, BMI, glucose, total cholesterol, HDL-C, LDL-C, triglycerides, and smoking habit. 

Considering that the prevalence of T2DM (35%) and hypertension (57%) are highest in ACS patients versus healthy controls (10% and 29%, respectively), we performed a sub-analysis of the polymorphisms associated with a low risk of ACS (*SELP* Ser290Asn and *SELP* Thr715Pro). This analysis was made comparing individuals with and without T2DM and the other hand, individuals with and without hypertension. The results show that both polymorphisms were not associated with T2DM or with hypertension (Appendix A). Therefore, this analysis corroborates that the genetic variation of these polymorphisms of the *SELP* gene are associated to the ACS, and not comes of the T2DM or hypertension.

### 3.3. Linkage Disequilibrium Analysis

We used the Haploview version 4.1 program for the analysis of the linkage disequilibrium and construction of haplotypes. In this analysis, the *SELP* Thr715Pro and *SELP* Leu599Val SNPs showed a strong linkage disequilibrium (D’ = 0.95). In addition, Haploview revealed strong evidence of recombination of the polymorphisms *SELP* Thr715Pro versus *SELP* Ser290Asn and *SELP Leu599Val* versus *SELP* Ser290Asn (D '= 0.17 and D' = 0.28, respectively; data not shown). This analysis marked three haplotypes with different distributions in ACS patients and healthy controls (Table 3). The “Thr-Leu-Ser” haplotype was associated with a higher risk of developing ACS (OR = 1.28, 95% CI: 1.05–1.54, pC = 0.006), while the *“*Pro-Leu-Ser” and “Thr-Leu-Asn” haplotypes were associated with a lower risk of developing ACS (OR = 0.72, 95% CI: 0.52-0.99, pC = 0.022, and OR = 0.71, 95% CI: 0.51–1.00, pC = 0.027, respectively).

### 3.4. Association of Polymorphisms with Plasma P-Selectin Levels 

In order to define the functional effect of the *SELP* Ser290Asn and *SELP* Thr715Pro SNPs associated with a lower risk of ACS, we determined the plasma levels of P-selectin in individuals with different genotypes of these two polymorphisms. For this analysis, we included a subgroup of 30 healthy controls for *SELP* Ser290Asn (7 *AA*, 11 *GA* and 12 *GG*) and a subgroup of 30 healthy controls for the *SELP* Thr715Pro SNP (8 *CC*, 11 *AC* and 11 *AA*). In this study, we did not include the analysis of plasma P-selectin levels in patients with ACS, due to the fact that in the setting of the coronary syndrome, the comorbidities, such as insulin resistance/T2DM, hypertension, and inflammatory processes, as well as the use of the anti-dyslipidemic and/or anti-hypertensive drugs, may have altered the inflammatory markers levels, such as inflammatory cytokines, adhesion molecules, and C-reactive protein, masking the real impact of *SELP* polymorphisms on plasma P-selectin levels [23,24,25]. In this context, subjects carrying the *AA* (Ans/Ans*)* genotype of the *SELP* Ser290Asn SNP had a lower P-selectin plasma concentration (33.93 ± 9.79 ng/mL) than carriers of the *GG (*Ser/Ser*)* (44.76 ± 6.54 ng/mL, *p* =0.032) or GA *(*Ser/Ans*)* genotypes (48.04 ± 16.57 ng/mL, *p* = 0.049) (Figure 1A). On the other hand, the analysis of the *SELP* Thr715Pro polymorphism showed that individuals with the *CC* (Pro/Pro) genotype had a lower concentration of P-selectin (26.44 ± 10.77 ng/mL) than *AA* (Thr/Thr) carriers (55.35 ± 14.05 ng/dl, *p* = 0.001). In addition, the individuals with the *AC (*Thr/Pro*)* genotype had lower P-selectin levels than *AA* (Thr/Thr) carriers (34.91 ± 14.46 ng/dl, *p* = 0.005) (Figure 1B).

### 3.5. Functional Prediction 

The functional prediction analysis showed that the presence of the *A (*Asn*)* allele of the *SELP* Ser290Asn polymorphism potentially produces a binding motif for Srp40 protein. In contrast, no evidence of potentially functional motifs was found for the *SELP* Thr715Pro polymorphism.

## 4. Discussion

In this study, we analyzed three relevant polymorphisms (Ser290Asn, Leu599Val, and Thr715Pro, respectively) of the *SELP* gene. The association of these SNPs with several inflammatory diseases in different populations is controversial, with positive and negative results [8,9,10,11,12,13,14,15]. In our study, the distribution of the *SELP* Leu599Val SNP was similar in both ACS patients and healthy controls. Nonetheless, the presence of the 290*Asn* and 715*Pro* alleles (*SELP* Ser290Asn and *SELP* Thr715Pro polymorphisms, respectively) was associated with a lower risk of developing ACS. In the same way, Reiner et al. reported in the CARDIA study that the *SELP* Ser290Asn and *SELP* Thr715Pro SNPs are associated with carotid intima-media thickness in young adults; however, these associations are different in European-American and African-American individuals [9]. In line with these data, Nasibullin et al. reported that the 290*Ans* allele of the *SELP* Ser290Asn SNP is associated with a lower risk of MI in a Russian population [13]. Similarly, the study of the risk of atherosclerosis in communities (ARIC), as well as the study of the Framingham heart (FHS) have shown that the genotype Pro715Pro is associated with a decreased risk of atherosclerosis in American and European populations [26,27]. In contrast with these data, in the ARIC study, Volcik et al. reported that the 290*Ans* and 715*Pro* alleles (*SELP* Ser290Asn and *SELP* Thr715Pro SNPs, respectively) were associated with the development of coronary heart disease in white but not in African Americans [11]. Similarly, Timasheva et al. reported that the *290*Ans allele of the *SELP* Ser290Asn SNP is associated with the development of hypertension in ethnic Tatars originating from the Republic of Bashkortostan (Russian Federation) [12]. By the same token, Kou et al. reported that *Thr715Pro* or *Pro715Pro* genotypes of the *SELP* Thr715Pro polymorphism increased the risk of developing cardiovascular diseases (CVD) in a Chinese Han population [28]. Additionally, we found that the H3 (Pro-Leu-Ser*)* and H4 *(*Thr-Leu-Asn*)* haplotypes were associated with a lower risk of developing ACS, whereas H1 (Thr-Leu-Ser) was associated with a higher risk. As can be seen, the haplotypic combinations between *SELP* Thr715Pro and *SELP* Ser290Asn polymorphisms were not in linkage disequilibrium. Nonetheless, the protection haplotypes carry *715Pro* and 290*Ans* alleles, and both of them were associated independently with a lower risk of cardiovascular diseases and other inflammatory diseases. This finding corroborated the role of these two alleles with the presence of ACS, whether they were analyzed independently or as haplotypes.

It is important to note that ACS patients and the healthy donors have much greater variation in blood glucose (102-188 versus 84-99) and diabetes mellitus (35% versus 10%). Considering these data, it is important to establish whether the polymorphisms are associated with T2DM or hypertension. In a sub-analysis, we showed that both polymorphisms were not associated with T2DM or with hypertension.

As can be seen, the associations of the *SELP* Ser290Asn and *SELP* Thr715Pro polymorphisms with ACS are contradictory in different study populations. We suggest that these discrepancies could be due to the classical cardiovascular risk factors and the environmental factors, such as diet, exercise, and lifestyle, which have an important role in the development of inflammatory diseases [29,30]. Another reason may be the fact that the allelic distribution of these polymorphisms varies according to the ethnic origin of the study populations. According to data obtained from the National Center for Biotechnology Information, populations from European, Asian, and African ancestry in Southwest US present a higher frequency of the *A* allele of the *SELP G1057A* Ser290Asn (rs6131) polymorphism (21.7%, 20.2% and 32.9%, respectively) when compared to Mexican mestizos and white American populations with a lower frequency of the *A* allele (9% and 14%, respectively). Concerning the *SELP A2331C* Thr715Pro (rs6136) SNP, Mexican mestizos, Europeans, and white Americans present a higher frequency of the *C* allele (8%, 8.8%, and 8.2%, respectively) than populations with Asian and African ancestry (0.2% and 2.5%, respectively) (https://www.ncbi.nlm.nih.gov/variation/tools/1000genomes/), (https://www.ensembl.org/index.html). 

We further determined the effect of the *SELP* gene polymorphisms on plasma P-selectin levels using genotype groups. We found that the *AA* (290 Asn/Asn) and *CC* (715 Pro/Pro) genotypes were associated with low P-selectin levels. As far as we know, this is the first study that showed the association of the *SELP* Ser290Asn and *SELP* Thr715Pro polymorphisms in P-selectin levels in individuals without the use of the anti-dyslipidemic or anti-hypertensive drugs. These drugs may modify the levels of the inflammatory markers, such as pro-inflammatory cytokines, adhesion molecules and C-reactive protein, masking the real impact of *SELP* gene polymorphisms on plasma P-selectin [23,24,25]. Nonetheless, the results concerning the association between P-selectin plasma levels and heart diseases are still contradictory. For example, Reiner et al. reported in the CARDIA study that the *A* (290Asn) and *C* (715Pro) alleles are associated with decreased plasma P-selectin levels and with the risk of developing atherosclerosis [9]. By the same token, Volcik et al. documented that the 715Pro allele is associated with lower P-selectin levels in the Atherosclerosis Risk in Communities (ARIC) study [27]. Similarly, Lee et al. determined that the lower serum levels of P-selectin decreased the risk of atherosclerosis [26]. At the same time, other reports have shown that the 715Pro *(C)* allele increased the expression of SELP mRNA, as well as the concentration of P-selectin levels in other inflammatory diseases, such as rheumatoid arthritis and T2DM [8,31]. As far as we know, the precise mechanism by which low and/or high P-selectin levels are associated with ACS remains to be elucidated. Nonetheless, recent data provide evidence that P-selectin upregulation on the endothelial cell surface mediates the effects of angiotensin II (Ang II), which has an important role in the development atherosclerosis [32]. In addition, Ang II stimulates not only the production of several molecules (adhesion molecules, chemokines, and cytokines) but also the oxidation and uptake of LDL, which promotes endothelial dysfunction [6,32]. On the other hand, Ang II triggers the synthesis of matrix metalloproteinases, the plasminogen activator inhibitor-1, and the proliferation of vascular smooth cells; this effect leads to the destabilization of atherosclerotic plaques [6]. Furthermore, using bioinformatics tools, we determined the potential effect of the *SELP* gene polymorphisms associated with ACS. The analysis of the *SELP* Thr715Pro polymorphism did not provide evidence of potential functional motifs. Nonetheless, the analysis of the *SELP* Ser290Asn polymorphism showed that the 290 Asn (*A)* allele generates a binding site for the Srp40 proteins. These proteins have multiple functions in the pre-mRNA splicing process, as well as in the regulation of alternative splicing, which leads to the production of protein isoforms [33,34]. In this context, we think that future investigations are warranted to understand the effect of these polymorphisms on P-selectin levels. 

Some limitations should be considered. The P-selectin levels were only measured in a small sample of control individuals and experiments on RNA transcription or protein stability were not made. Considering these limitations, the effect of the SNPs on P-selectin plasma levels should be taken with care and studies in a large number of individuals are necessary to corroborate this association. In the same way, in our study it was not possible to determine the expression levels of P-selectin on the leukocyte’s surface to confirm the data obtained in plasma. 

In summary, this study demonstrated that the *SELP* Ser290Asn and *SELP* Thr715Pro polymorphisms are associated with a lower risk of developing ACS in a Mexican population. It was possible to distinguish two haplotypes (Pro-Leu-Ser and Thr-Leu-Asn) associated with a lower risk of developing ACS. On the other hand, both polymorphisms were associated with lower P-selectin levels in plasma. Lastly, due to the specific genetic characteristics of the Mexican population, we consider that additional studies will need to be undertaken in a larger number of individuals and in populations with different ethnic origins; these studies could help define the true role of these polymorphisms as markers of risk or protection from developing ACS and other cardiovascular events.

## Figures and Tables

**Figure 1 biomolecules-10-00270-f001:**
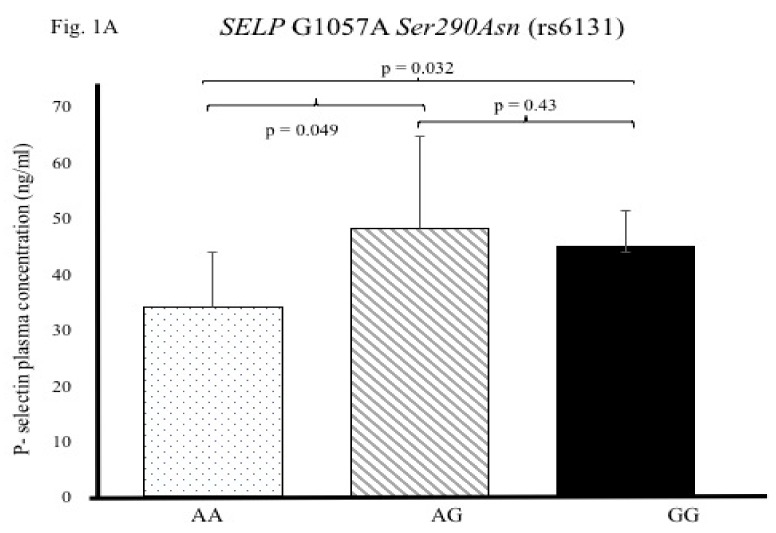
Genetic contribution of the *SELP G1057A* and *SELP A2331C* polymorphisms on P-selectin levels. (**A**) P-selectin plasma levels in individuals with different genotypes of the *SELP G1057A* polymorphism. (**B**) P-selectin plasma levels in individuals with different genotypes of the *SELP A2331C* polymorphism.

**Table 1 biomolecules-10-00270-t001:** Clinical characteristics and biochemical parameters of the study individuals.

		ACS (*n* = 625)	Healthy Controls (*n* = 700)	*p*-Value
		Median (percentile 25–75)	Median (percentile 25–75)	
Age (years)		57.72 (51–65)	54.39 (49–59)	<0.001
BMI (kg/m^2^)		27.3 (25–29)	28.3 (26–31)	0.001
Blood pressure (mmHg)	Systolic	130.61 (114–144)	117.32 (106–126)	<0.001
	Diastolic	80.1 (70–90)	72.47 (66–77)	<0.001
Glucose (mg/dl)		158.51 (102–188)	98.73 (84–99)	<0.001
Total cholesterol (mg/dl)		164.22 (128–198)	190.4 (164–210)	<0.001
HDL-C (mg/dl)		38.32 (32–44)	44.6 (35–53)	<0.001
LDL-C (mg/dl)		106.4 (76–133)	115.8 (94–134)	<0.001
Triglycerides (mg/dl)		169.2 (109–201)	175.1 (112–208)	0.218
Gender n (%)	Male	510 (82)	463 (66)	<0.001
	Female	115 (18)	237 (34)	
Smoking n (%)	Yes	225 (35)	155 (22)	<0.001
Hypertension	Yes	355 (57)	206 (29)	<0.001
Diabetes mellitus	Yes	218 (35)	68 (10)	<0.001
Dyslipidemia n (%)	Yes	534 (85)	501 (71)	<0.001

Data are expressed as median and percentiles (25th–75th). *p-*values were estimated using Mann–Whitney U test for continuous variables and chi-square test for categorical values. ACS: acute coronary syndrome patients.

**Table 2 biomolecules-10-00270-t002:** Distribution of *SEL-P* polymorphisms in ACS patients and healthy controls.

				MAF	Model	OR (95%CI)	*pC*
***SELP G1057A*** **Ser290Asn (rs6131)**							
	GG	GA	AA				
Control (*n* = 691)	569 (0.823)	115 (0.166)	7 (0.010)	0.09	Co-dominant Dominant Recessive	0.59 (0.38–0.92) 0.59 (0.39–0.90) 0.58 (0.12–2.83)	0.047 0.014 0.49
ACS (*n* = 617)	541 (0.877)	73 (0.118)	3 (0.005)	0.06	Over-dominant Additive	0.61 (0.39–0.92) 0.62 (0.42–0.92)	0.019 0.015
***SELP G1980T*** **Leu599Val (rs6133)**							
	GG	GT	TT				
Control (*n* = 682)	563 (0.825)	114 (0.167)	5 (0.007)	0.09	Co-dominant Dominant Recessive	0.28 (0.02–3.32) 1.07 (0.73–1.58) 0.27 (0.02–3.26)	0.46 0.73 0.26
ACS (*n* = 611)	505 (0.827)	105 (0.172)	1 (0.002)	0.09	Over-dominant Additive	1.12 (0.75–1.66) 1.02 (0.71–1.49)	0.57 0.90
***SELP A2331C Thr715Pro*** **(rs6136)**							
	AA	AC	CC				
Control (*n* = 685)	580 (0.847)	97 (0.141)	8 (0.012)	0.08	Co-dominant Dominant Recessive	0.63 (0.40–0.99) 0.61 (0.39–0.95) 0.36 (0.04–2.95)	0.075 0.028 0.32
ACS (*n* = 607)	537 (0.884)	67 (0.110)	3 (0.005)	0.06	Over-dominant Additive	0.63 (0.40–0.99) 0.62 (0.41–0.94)	0.044 0.023

ACS, acute coronary syndrome, MAF, minor allele frequency, OR, odds ratio, CI, confidence interval, *pC*, *p*-value corrected. The p-values were calculated by the logistic regression analysis, and the ORs were adjusted for gender, age, blood pressure, BMI, glucose, total cholesterol, HDL-C, LDL-C, triglycerides, and smoking habit.

**Table 3 biomolecules-10-00270-t003:** Haplotype frequencies (Hf) of *SEL-P* haplotypes in ACS patients and healthy controls.

Haplotypes	Thr715Pro	Leu599Val	Ser290Asn	ACS (n = 605)	Controls (n = 676)	OR	95%CI	P
				Hf	Hf			
H1	Thr	Leu	Ser	0.804	0.763	1.28	1.05–1.54	0.006
H2	Thr	Val	Ser	0.073	0.067	1.08	0.80–1.47	0.32
H3	Pro	Leu	Ser	0.057	0.077	0.72	0.52–0.99	0.022
H4	Thr	Leu	Asn	0.049	0.066	0.71	0.51–1.00	0.027
H5	Pro	Val	Asn	0.014	0.022	0.62	0.33–1.13	0.063

Abbreviations: Hf, Haplotype frequency; *P*, *p*-value; OR, odds ratio; 95% CI, confidential interval. The order of the polymorphisms in the haplotypes is according to the positions in the chromosome (*SELP A2331C Thr715Pro* (rs6136), *SELP G1980T* Leu599Val (rs6133), and *SELP G1057A* Ser290Asn (rs6131). Bold numbers indicate significant associations.

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
