# Peer review of "The Ser290Asn and Thr715Pro Polymorphisms of the SELP Gene Are Associated with A Lower Risk of Developing Acute Coronary Syndrome and Low Soluble P-Selectin Levels in A Mexican Population â€"

_biomolecules, 2020, doi:10.3390/biom10020270_

Round 1
Reviewer 1 Report
Based on the cases study, Gabriel et al. tried to characterize the impact of the P-selectin gene polymorphism on acute coronary syndrome. The authors compared the distribution of SELP gene polymorphisms in ACS patients and healthy people, and claimed an association between these SNPs and the risk of ACS.
1, As shown in Table 1, the ACS patients and the healthy donors has much greater variation in blood glucose (102-188 versus 84-99) and diabetes mellitus (55% versus 10%). The authors should discuss this in the manuscript.
2, The authors claimed that the SNPs is associate with the risk of ACS. However, according to Table 2, the SNP variation of ACS and Control is very close. Therefore, as mentioned in my first question, it's important to know whether the variation comes from the diabetes mellitus or hypertension? The authors should further analyze their data to demonstrate this.
3, Direct evidence that SNPs direct contribute to ACS is lacking as the ACS patient P-selectin level is unknown.
4, And while the authors concluded that the SNPs has effects in plasma P-selectin level in a few donor samples, a direct evidence is lacking. Can the authors show that these SNPs direct affect RNA transcription or protein stability when expressing these mutants in cell lines?
5, Rather than compare plasma P-selectin levels, the authors may also want to check the expression of P-selectin protein level on the leukocytes surface to confirm their findings.
6, The authors need to provide more evidence/discussion on how levels of P-selectin contribute to ACS.
Author Response
"Please see the attachment"

Reviewer 2 Report
Herrera-Maya et al. in the original research manuscript entitled “The Ser290Asn and Thr715Pro polymorphisms of the SELP gene are associated with a lower risk of developing acute coronary syndrome and low soluble P-selectin levels in a Mexican population” investigated the association of three single nucleotide polymorphisms (SNP) with the incidence of acute coronary syndrome (ACS) and plasma P-selectin levels. The authors observed a decreased risk of ACS with GA and AC genotypes of Ser290Asn and Thr715Pro SNP in SELP gene. Furthermore, they found reduced plasma P-selectin levels in individuals with these genotypes. The manuscript is well-written and easy to follow. They studied a large cohort of individuals. However, I have the following minor concern.
It would have been better if the authors have performed sequencing of DNA fragments consisting these SNPs for a few samples to confirm the results obtained with the TaqMan assay.
Author Response
"Please see the attachment"

Round 2
Reviewer 1 Report
My concerns have been addressed.